# Reducing cost uncertainty in the drivetrain design decision with a focus on the operational phase

Freia Harzendorf[1], Prof. Ralf Schelenz[1], Prof. Georg Jacobs[1]

[1]Chair for Wind Power Drives, RWTH Aachen University, Aachen, 52074, Germany

*Correspondence to*: Freia Harzendorf (freia.harzendorf@rwth-aachen.de)

**Abstract**

In order to identify holistically better drivetrain concepts for onshore application, their operational behaviour needs to be considered in an early design phase. In this paper, a validated approach for estimating drivetrain concept-specific unplanned operational effort and risk based on open access data is presented. Uncertain influencing factors are described with distribution

functions. This way, the poor data availability in the early design phase can be used to give an indication about the concept's choice influence on the unplanned operational turbine behaviour. In order to get representative comparisons, a Monte Carlo method is applied. Technical availability and drivetrain influenced unplanned operational effort are defined as evaluation criteria. The latter is constituted by labour, material, and equipment expenses. By calculating the range of fluctuation of the evaluation criteria mean values, this approach offers an indication about the inherent risk in the operational phase induced by

the drivetrain concept choice.

This approach demonstrates that open access data or expert estimations are sufficient for comparing different drivetrain concepts over the operational phase in an early design stage when using the right methodology. The approach is applied on the five most common state-of-the-art drivetrain concepts. The comparison shows that among those concepts the drivetrain concept without a gearbox and with a permanent magnet synchronous generator performs the best in terms of absolute drivetrain

influenced unplanned operational effort over the lifetime as well as on the inherent risk for the assumptions made. It furthermore makes it possible to give insights on how the different drivetrain concepts might perform in future applications in terms of unplanned operational effort. Exemplary the impact of higher torque density in gearboxes, a change to moment bearings and adjusted coil design in electrically excited generators have been analysed. It shows, that the superiority of synchronous generator concepts manifested in historic data is not entirely certain in future applications. Concluding, this

approach will help to identify holistically better drivetrain concepts by being able to estimate the inherent risks and effort in the operational phase.

## 1. Introduction

Decreasing subsidies, fierce competition to fossil power stations and photovoltaics puts the wind industry under high development and cost pressure. The wind turbine drivetrain as the sum of the energy converting components between hub and transformer has a significant influence on the turbine's properties and behaviour. Up to 50 % of the turbines investment cost can be accounted by the nacelle and its components (Mone et al., 2015). More importantly over 80 % of the unplanned failures of a turbine can be traced back to nacelle components (Reder et al., 2016). It is estimated that cost arising during the operational phase can add up to the initial investment cost (Luers et al., 2015).

Today's market presents a variety of drivetrain concepts. However, no statement about the best concept is yet possible. Especially the concept's performance in the operational phase is hard to estimate upfront. Its components are designed for a 20 year lifetime with not fully known load cases, maintenance and mounting accuracy. These are especially not known during the conceptualization phase. Though, the greatest influence on the product's, in this case the drivetrain's, success can be exerted in the early phases of product development as its cornerstones are set here (Ehrlenspiel et al., 2014). Furthermore, effort for design modification rises exponentially with the products maturity level (Ehrlenspiel et al., 2014). To identify superior products in an early phase of the product development, this paper argues that a concept-specific estimate about the unplanned maintenance effort and inherent risks is required. This can be done in a two-step approach. First of all, the drivetrain concept characteristics which have an influence on the operational phase of an onshore wind turbine have to be identified. This includes the description of their influence. Subsequently a method for modelling this behaviour in an early design stage needs to be identified and implemented. This paper aims at providing information about the expected drivetrain component and concept operational behaviour as well as a statement about the certainty of this behaviour. The outcomes of this paper provide a turbine designer with a tool to identify holistically better drivetrain concepts for onshore application and furthermore evaluate possible places for improvement and its influence on the operational phase.

In the following, an approach for estimating drivetrain concept-specific inherent risk of unplanned maintenance effort and technical availability is developed and presented. In Section 2, a short literature review is given. Section 3 presents the paper's object of reflection. Section 4 introduces the general model approach. In Section 5, the developed model and its underlying assumptions are introduced. The required validation is stated in Section 6 and in Section 7, a concept comparison is conducted. Finally, Section 0 gives a conclusion and an outlook.

## 2. Literature review

The evaluation of the unplanned operational behaviour of wind turbine drivetrains in an early design phase is rare. Nevertheless, some drivetrain concept comparisons focusing on the operational phase are available in literature. Most of them derive statements based on the evaluation of empirical databases, which are unfortunately not open access (Carroll et al., 2014b). Thereby, Failure Mode Effect Analysis and derivates (Cevasco et al., 2018; Ozturk et al., 2018) as well as Monte Carlo simulations (McMillan and Ault, G.,W., 2010; Dalgic et al., 2015) are the most commonly used methods. They highly

rely on empirical databases which are not available in the early phase of product development. Alternatively, other authors use
fixed average failure rates from one source to model the components operational behaviour (Carroll et al., 2014a; Carroll et
al., 2017). As shown by Carroll et al. 2015 the representativeness of analysis based on fixed average failure rates from one
source is questionable (Carroll et al., 2015b). In addition, available concept comparisons mostly lack an indication about the
certainty of their results. In an uncertain situation it helps to at least have an indication about the level of uncertainty and its
source(s). Furthermore, the above presented approaches are not suitable for a technology comparison if aiming for identifying
room for improvement from a technological perspective. Statements about the lifetime behaviour as well as scalability are
mostly not in the scope. Hence this publication presents an approach for deriving scalable and more representative estimations
about a drivetrains concept specific operational behaviour based on publicly available data.

### 3.   Object of reflection

This paper aims to quantify the influence of the drivetrain concept choice on the operational expenditures over the turbine's
lifetime as well as on the turbine's technical availability. The focus lies on the consideration and quantification of uncertain
aspects of unplanned operational effort. In this approach, drivetrain is seen as the sum of the energy converting components
between the turbine's hub and transformer. This means the operational behaviour of the chosen suspension system, gearbox,
generator and converter design are considered. Figure 1 gives an overview about aspects generally influencing the operational
expenditures of a turbine. They are divided into aspects being directly influenced by the drivetrain concepts choice as well as
aspects being uncertain.

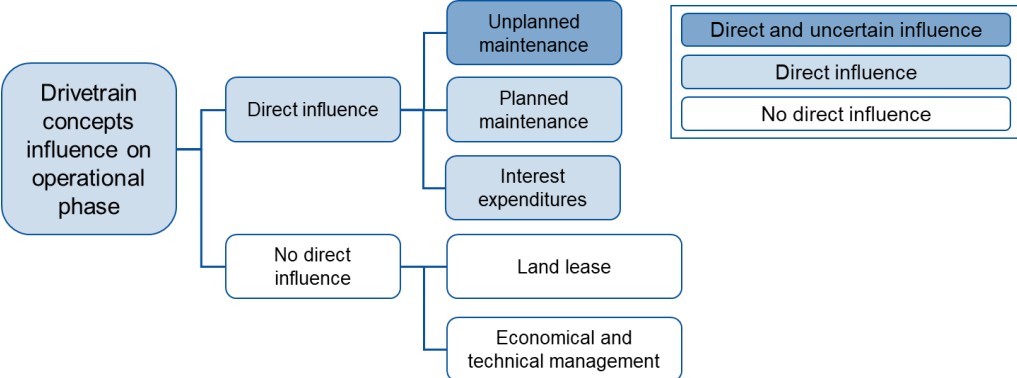

**Figure 1: Factors influencing the operational phase of a wind turbine**

The focus of this investigation lies on drivetrain influenced uncertain aspects. Unplanned maintenance is the most prominent
factor which is uncertain and directly influenced by the concept choice. Therefore, it is solely considered in this approach.
Unplanned maintenance is defined as an unpredictable component breakdown which urgently needs unscheduled activities.
The unpredictable component breakdown makes this aspect a highly uncertain and risk inherent situation. It can have a
multitude of influencing factors like the component design, unknown loading conditions, system interaction, manufacturing

and mounting accuracy. When trying to find help in literature, data is often anonymized and therefore samples cannot be characterized in a sufficient way (Cevasco et al., 2018). More importantly different studies come to contradictory statements about the components failure behaviour (Ozturk et al., 2018; Carroll et al., 2017). Unplanned maintenance includes unscheduled activities that need to take place in case of a component breakdown. The needed actions and the related effort are mostly uncertain and again are influenced by a multitude of factors. Failure type, accessibility, weather, spare part, technicians and equipment availability can influence the unscheduled activities. Once more, literature studies seldomly provide information about durations (downtimes, repair times) and reasons for the extent of the activities. Samples are defined in an unsatisfactory way. These complex and uncertain features make it impossible to precisely calculate the unplanned maintenance effort and availability of a drivetrain concept in an early design phase with respectable effort. Still this is a major characteristic of a drivetrain concept which has to be considered in the concept decision.

In this paper the early design phase is defined as the phase in the product development process where design decisions for the concept are made (cf. step three of VDI 2221 (VDI, 1993)). This phase is characterized by a high degree of complexity, uncertainty and information deficits. In the status quo, this highly important decision is mainly based on experience of the deciding engineers. This can be especially critical when evaluating completely new ideas differing to the former product generation. Known in this decision are the rated power of the turbine, its rotor diameter, the wind class it is developed for and the possible drivetrain concepts.

## 4. Model approach

This section presents the used approach for estimating drivetrain concept-specific unplanned maintenance effort and technical availability in an early design stage. The approach needs to fulfil the following requirements:

- Deal with the poor availability of concept-specific information in literature and early design stage
- Allow estimates about the technologically inherent impact the drivetrain concepts choice has on the operational phase
- Consider and evaluate the most relevant influencing factors in the operational phase
- Be applicable to state-of-the-art drivetrain concepts
- Be scalable in rated power and rotor diameter
- Be applicable to incremental inventions and new concept ideas

This approach is based on publicly available studies about the drivetrains' operational phase. As mentioned in Section 3, these studies sometimes come to contradictory statements and are not always transparent about the cause of failure or downtime. Therefore, the model is based on several assumptions. The first assumption is that not all influencing factors leading to a failure can be modelled individually. Therefore, failure detectability, weather or site-specific impacts as well as the maintenance strategy itself are not considered directly. Furthermore, it is assumed, that all available study results from literature represent realistic component behaviour, as it is mostly not known what conditions the underlying turbines experienced or which specific failure mechanism occurred. The next assumption is, that this behaviour is mainly influenced by technological choice. It is

assumed that these influencing factors are randomly distributed. Failure rate, downtime, failure severity, and duration of repair and replacement are modelled as uncertain factors. In order to include all available information, continuous distributions are chosen to fit the observed data for depicting the uncertain factors if possible. Parameters for fitting the distributions are estimated based on a maximum likelihood method. It is assumed that the entire drivetrain consists of repairable assemblies, which means each assembly can sustain more than one failure and is 'as good as new' after repair or replacement. In reality, repair never reaches the reliability of a new component. Still, this assumption makes it possible to model the life of a fictional wind turbine based on the derived distributions.

A statistical approach, the Monte Carlo method, is utilized for deriving representative results as it makes it possible to calculate a multitude of fictional turbine lives. It has the ability to conduct a high number of random experiments based on uncertain influencing variables. The basis for this method is the law of large numbers. It says that, by performing a large number of experiments, the mean of the results will get close to the expected value. This approach is suitable for the present problem as it is constituted by different uncertain factors that can be described by continuous distribution functions. Furthermore, this method offers the possibility to not just get insights on the expected value but also about the result's occurrence probability. The inverse-Transform sampling method is used for generating random numbers with a defined distribution. This way, a sufficient number of fictional wind turbine operational lifetimes are simulated for every component based on the distributions derived from literature data. This is done for all relevant components.

In this approach, technical availability AV [%] is influenced by uncertain factors including mean time to failure and duration of repair, replacement or downtime, c.f. Formula (1):

$$AV(i) = \frac{\sum_{s=1}^{3} \sum_{j=1}^{4} \sum_{d=1}^{4} do_{j,d,s} * f_{j,d,s}(i)}{h} \tag{1}$$

In this formula, $j$ indicates the component (main bearing, gearbox, generator, converter), $d$ the specific design (e.g. for generator PMSG) of the component, and $s$ the failure severity (minor repair, major repair or major replacement). The amount of failures in the specific year $i$ is represented by $f [\frac{failure}{a}]$. The downtime each failure leads to is represented by $do [\frac{h}{failure}]$ in year $i$. The variable $h$ is used to show the number of hours a calendar year has $[8760 \frac{h}{a}]$. Technical availability is therefore calculated as the percentage of the years' time where the drivetrain could technically provide electricity if wind conditions are met.

Estimating the drivetrain influenced unplanned operational effort (DUOE) [€] is a bit more complex cf. Formula (2):

$$DUOE(i) = \sum_{s=1}^{3} \sum_{j=1}^{4} \sum_{d=1}^{4} f_{j,d,s}(i) * (LE(dr_{j,d,s}, nt_s, w) + ME(m_{j,d,s}) + EE(c(we_{j,d,s}, dr_{j,d,s}))) \tag{2}$$

It is constituted by labour, material and equipment expenses. Labour expenses $LE$ [€] are influenced by the uncertain factor duration of repair or replacement $dr$ [h], which is component, design and failure severity dependent. $LE$ is furthermore influenced by the number of needed technicians $nt$ [-], which is failure severity dependent. Finally, the wage of a technician $w$ [€/h] impacts the labour expenses. Material expenses $ME$ [€] are determined taking the severity of the failure and component design specific investment cost m [€] into account. Equipment expenses $EE$ [€] consider expenses for a crane to enable

component exchange. The needed crane and its associated expenses c [€] are dependent from the component design specific

weights we [kg] and the duration of repair or replacement $dr$ [h].

Both component design specific weight and component design specific investment cost scale with rated power and rotor diameter and therefore with the field of application. They are calculated based on the NREL Cost and Scaling Model (Fingersh et al., 2006), which is a cost and mass regression model based on industry data. As visible in Formula (2) these two variables have an impact on the material expenses as well as on the equipment expenses leading to a high impact on DUOE. Therefore,

the use of these inputs makes this approach scalable in rated power and rotor diameter.

## 5. Model implementation

The following section gives insights on how the model idea is implemented. Some general assumptions are presented in the beginning before the model procedure is introduced. Failure rate, downtime, failure severity, duration of repair and replacement are modelled as uncertain factors. Collected data about these factors is allocated to the different components and their design.

Design unspecific information is assorted to the component in general. This unspecific information is later considered for all component designs. This allows to make the most out of the available data while not favouring one design or distorting the result. Figure 2 shows an overview of the models' structure and the underlying assumptions. Model input is constituted by the component design, rated power, and rotor diameter. One model iteration represents the operational behaviour of a drivetrain from installation until end of its design lifetime.

**Table 1: Failure severity distinction based on Carroll and model implementation (Carroll et al., 2014b)**

| Failure severity distinction | Minor repair | Major repair | Major replacement |
|---|---|---|---|
| Definition (Carroll et al., 2014b) | Material cost up to 1,000 € | Material cost between 1,000 € to 10,000 € | Material cost over 10,000 € |
| Material expenses | 0 | Random number between 1,000 € – 10,000 € | Component investment cost |
| Labour expenses | f (repair time) | f (repair time) | f (replacement time) |
| Equipment expenses | - | - | Additional crane |

For every operational year component failure occurrence and failure time are calculated. It is assumed that the components failure behaviour follows a Weibull distribution. This is a common assumption for technical systems. Weibull distribution makes it possible to reveal the main nature of the failure being premature, random, or due to wear out. Weibull parameters for the failure behaviour of the different components are determined based on mean time to failure. Mean time to failure as the

reciprocal of failure rates is derived from available failure rates from literature (for sources see table in Figure 2). Unfortunately, the sample cannot be characterized completely. The sample is mainly constituted by data recorded in between 1990 - 2014 for a rated power up to 4 MW. A maximum likelihood method is applied for deriving the Weibull parameters for mean time to failure. It is assumed that failure rates for the different component designs already contain subsequent faults due to the chosen system. Therefore, components can be modelled independently from each other.

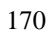

Figure 2 diagram content:

**Legend**
- Results
- Calculated model inputs

*while $t \leq t_{design}$*

**INPUT**
- Components design
- Rated power
- Rotor diameter

**Concept specific mean time to failure**
**Weibull distribution/ Triangulation**

Failure

**NREL Cost and Scaling Model**
- Weight
- Investment cost

**Crane model**
- Crane cost

Literature failure rates

Literature

**Downtime**
**Normal distribution**

**Severity of failure**
**Uniformly distributed random number**
**(Carroll et al. 2015b)**

no

yes

**Operational expenditures**
- Material cost
- Labour cost
- Crane cost

**Estimation of technical avialability (AV)**

**Duration of repair**
**Triangulation**

Literature

**Estimation of O & M effort (OME)**


| Uncertain factor | Model implementation | Source |
|---|---|---|
| Failure rate | Weibull distribution/Triangulation | (Fischer and Wenske, 2015; Fischer et al., 2015; Ozturk et al., 2018; Shafiee and Dinmohammadi, 2014; Ribrant, 2006; Dinmohammadi and Shaffiee, 2013; Arabian-Hoseynabadi et al., 2010; Carroll et al., 2017; Berger, 2016; Reder et al., 2016; Carroll et al., 2016; Dinwoodie and McMillan, 2012; Pinar P., J., M. et al., 2013; Wilson and McMillan, 2014; Carroll et al., 2014a; Tavner and Spinato, 2008) |
| Downtime | Normal distribution | (Fischer and Wenske, 2015; Samet Ozturk, Vasilis Fthenakis and Stefan Faulstich; Ribrant, 2006; Carroll et al., 2017; Reder et al., 2016; Dinwoodie and McMillan, 2012; Pinar P., J., M. et al., 2013; Carroll et al., 2016) |
| Failure severity | Uniformly distributed random number | (Carroll et al., 2015a) |
| Duration of repair | Triangulation/constant | (Carroll et al., 2016; Carroll et al., 2017) |

**Figure 2: Overview of model structure**

In case of a failure, its severity needs to be determined. Referring to Carroll et al., failure severity categorizes failures due to their impact on material cost (Carroll et al., 2014b). It is distinguished between minor repair, major repair, and major replacement. The first row in Table 1 gives the definition of the used failure severity types. Failure severity is considered with

a uniformly distributed random number and a percentual distribution determined from (Carroll et al., 2015b). Unfortunately, this distribution is deduced from an offshore database.

Failure severity affects the downtime of a turbine. For every failure severity category downtime is modelled in a distinctive way. Downtime due to minor repair is modelled with a constant value from literature on (Carroll et al., 2017). For major repair and replacement, downtime is assumed to follow a normal distribution. Distribution parameters are derived from literature

(please compare table in Figure 2). The accumulated downtime over the drivetrain's design lifetime now allows an estimate about the effect of the unplanned drivetrain failures on AV.

According to Formula (2) the estimation of DUOE is constituted by material, labour and equipment expenses (please compare Table 1). Material expenses estimation is described in the following paragraph. Minor repair is repair which leads to material cost up to 1,000 €. In this model material expenses are therefore neglected. Major repair is implemented as a random number

between 1,000 – 10,0000. According to Carroll major replacement is a replacement which leads to material cost exceeding 10,000 €. In the model it is assumed that the entire component needs to be exchanged if this failure type occurs. Material expenses are therefore modelled as the investment cost of the failed component. Component and design specific component investment cost is calculated based on rated power and rotor diameter using the NREL Cost and Scaling Model (Fingersh et al., 2006).

Labour expenses, another stake of DUOE, is mainly influenced by the duration of the action. Failure severity and component specific actions duration for major repair and replacement is modelled with the help of a triangulation. Here the modus is assumed to equal the mean. For minor repair a fixed action duration per component is taken into account based on (Carroll et al., 2017). Repair is done by two technicians. Replacement measures require three technicians due to safety reasons. A constant hourly wage is assumed.

Finally, equipment expenses need to be estimated. It is assumed, that no additional crane is needed for minor and major repair as the onboard equipment can be used. An additional crane is used to enable the component exchange for major replacements. For crane cost estimation, a parameterized model is developed which chooses the needed crane based on component weight and hub height the component needs to be lifted on. Crane data is based on Liebherr cranes (Liebherr). Component weight is estimated based on the NREL Cost and Scaling Modell (Fingersh et al., 2006). For some components an exchange is only

possible if further components are dismounted, this fact is considered in the crane decision. The crane is leased for the time the replacement takes.

By adding up the three expenses a concept specific estimate about DUOE is possible.

### 6. Model validation

Verification and validation are done by comparing modelled values with published data combined with a general reasonability

check. In the beginning, the failure behaviour is in the focus. Components in the following designs are in the scope: moment, trunnion, 3-point and 4-point suspension system, two and three-stage gearbox, permanently magnet synchronous generator

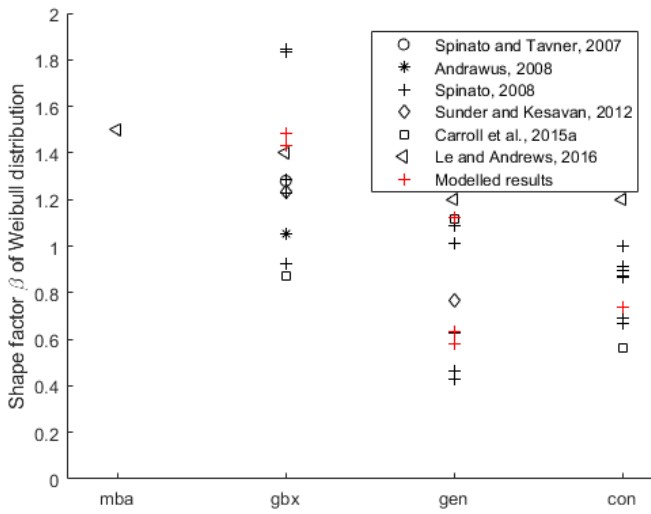

**Figure 3: Components failure behaviour from literature and model results described by the Weibull shape factor (Spinato and Tavner, 2007; Andrawus, 2008; Spinato, 2008; Sunder and Kesavan, 2012; Carroll et al., 2015b; Le and Andrews, 2016)**

(PMSG), electrically excited synchronous generator (EESG) and doubly fed induction generator (DFIG) as well as partially and fully rated converter. Initial null hypothesis is that all components failure behaviour can be described by a Weibull distribution. Due to the small sample size, an Anderson-Darling goodness of fit test is conducted. This test is applicable to samples with a minimum size of four. Null hypothesis for a Weibull distribution is not rejected for two stage gearbox and three stage gearbox with a three point suspension system, all generator types and partially rated converters with a five percent

significance level. Hence, they are modelled by a Weibull distribution. The three-stage gearbox with a four-point suspension system follows a log-normal distribution again confirmed by an Anderson-Darling goodness of fit test. For all main bearing arrangement designs as well as fully rated converters, this test is either not applicable or the null hypothesis is rejected. Therefore, a triangulation is applied. An Anderson-Darling goodness of fit test supports the assumption that components downtime can be described by a normal distribution. Unfortunately, no design specific modelling for downtime is possible due

to a lack of data.

There are a few publications available in literature where the failure behaviour of different wind turbine drivetrain sub-assemblies has been empirically evaluated and described by a Weibull distribution. Figure 3 shows the shape factor of the Weibull distribution for different components failure behaviour from literature and the modelled results. A first look reveals a widespread in the shape factor in literature not indicating an ambiguous failure behaviour. It needs to be considered that the

Weibull shape factors are not distinguished into the components design. Model results are component design specific and show different behaviour for the different designs which is in line with literature values. This way the chosen distributions and distribution parameters are confirmed.

Not only the failure behavior shall be validated but also the general model results, meaning the modelled mean DUOE and AV. The mean values are calculated based on results of 1,000,000 iterations. According to the law of large numbers, the

average results converge against the expected value the more iteration results are taken into account. Therefore, the calculated mean values can be seen as approximations of the expected value of DUOE on a lifetime basis. The expected value is the value which is the arithmetic mean and therefore the most probable outcome.

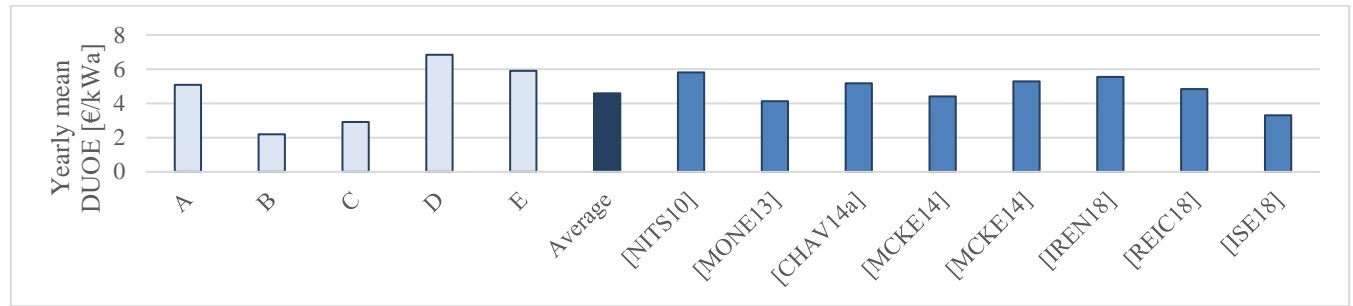

**Figure 4: Scaled Meta study about yearly mean DUOE and model results (Nitsch et al., 2010; ISE Fraunhofer, 2010, 2012, 2013;**
**Fichtner / Prognos, 2013)**

Literature does not directly provide these numbers needed for the comparison; therefore, they are approximated in the following. Yearly operational cost can vary between 2 – 4.2 % of the initial investment cost of the turbine (Nitsch et al., 2010; ISE Fraunhofer, 2010, 2012, 2013; Fichtner / Prognos, 2013). Two further created meta studies indicate that operational expenditures over the year vary from 30 to 52 €/kWa or 0.5 – 2.68 ct/kWh (Nitsch et al., 2010; Mone et al., 2013;
Chaviaropoulos and Natarajan, 2014; McKenna et al., 2014; IRENA - International Renewable Energy Agency, 2018; Reichenberg et al., 2018, 2018; Fraunhofer ISE, 2018). These meta studies give the impression that operational expenditures vary substantially. Unfortunately, the sources do not indicate their samples in a sufficient way. Therefore, only a scale comparison can be conducted for validation. 44 – 55 % of yearly operational expenditure is associated with maintenance and repair (Luers et al., 2015). For the comparison planned maintenance effort, and unplanned effort for other turbine components
needs to be excluded. This leads to the assumption that a quarter of the maintenance and repair expenses are caused by DUOE.

**Table 2: Considered drivetrain concept characteristics and their market share (Hernandez, 2017)**

| Concept | A | B | C | D | E |
|---|---|---|---|---|---|
| **Suspension system** | Moment | Moment | Trunnion | 3-point | 4-point |
| **Gearbox** | 2 stage | - | - | 3-stage | 3-stage |
| **Generator** | PMSG | PMSG | EESG | DFIG | DFIG |
| **Converter** | Fully rated | Fully rated | Fully rated | Partially rated | Partially rated |
| **Market share [%] (Hernandez, 2017)** | 10 | 35 | 10 | 40 | |

A corrected Meta study is shown in Figure 4. In addition to the literature values, this figure also depicts the calculated values for currently available drivetrain concepts (A – E). The concept characteristics including their market share are presented in Table 2. All concepts are designed for a rated power of 3 MW and a rotor diameter of 120 m, representing the currently
installed onshore fleet in Germany. This application will be used in all coming analysis in this paper if not stated differently. With an average yearly mean DUOE value of 4.59 €/kWa the modelled results are in between the meta study results varying

between 3.3 €/kWa and 5.808 €/kWa. Furthermore, the industry standard of a technical availability above 97 % is achieved for all analysed concepts. So, the general model results are reasonable.

## 7. Concept comparison

The validation section showed that there are significant differences in mean DUOE for different drivetrain concepts. This section allows to better understand underlying reasons for these differences. In a second step, the developed framework is used to address possible future developments and their impact on mean DUOE of different drivetrain concepts.

**Table 3: Modelled Weibull parameter for failure behaviour of different drivetrain components in different designs**

|  | 2-stage gbx | 3-stage gbx & 3-point suspension | PMSG | EESG | DFIG | Partially rated converter |
|---|---|---|---|---|---|---|
| **Shape factor** | 1.4311 | 1.4846 | 0.57877 | 0.63376 | 1.1203 | 0.73571 |
| **Scale factor** | 9.195 | 8.4066 | 14.5737 | 13.9907 | 9.3895 | 13.3346 |

First of all, the component design specific failure behavior is evaluated. Table 3 presents the Weibull parameters derived from
available historical literature sources (c.f. Figure 2) for the different components in their different designs. It is visible that the PMSG, EESG and partially rated converter mainly followed early failure behaviour in the past. They have a shape factor below 1. Statistically, one failure will occur during their lifetime as indicated by their scale factor. Whereas two-stage gearbox, three-stage gearbox with a three-point suspension system, and DFIG have mainly been attributed to wear out behaviour. For these component designs statistically two failures will occur over their lifetime, indicated by their scale factor below 10. In this
approach no specific failure mechanisms are discerned, they are aggregate into three failure types (premature, random or wear out). For future research the derivation of failure mechanism specific Weibull distributions could be a highly interesting topic. This way specific improvements in specific component designs could be directly taken into account.

Figure 5 gives an overview about the calculated mean DUOE over the entire drivetrains lifetime split into material, labour and equipment expenses share. All concepts and components have in common, that the material expenses have the highest influence
on mean DUOE followed by labour expenses. Equipment expenses, if modelled in the way presented, are less influential. Under the chosen assumptions direct drive concepts (B & C) lead to the lowest mean lifetime DUOE. To explain this, a deeper look onto different components in different designs is needed. Going from the bottom to the top of Figure 5 it starts at the component main bearing arrangement. No direct influence for the main bearing arrangement on the unplanned operational expenses is calculated. This can be explained by the mean time to failure used for the triangulation which is in the scale of $10^6$
years. Looking at the gearbox it is apparent, that it is the component being responsible for most of the unplanned operational expenditure of a drivetrain. This can be traced back to its failure behaviour which indicates statistically two failures occurring during its lifetime and 24 % of these failures leading to major replacement (Carroll et al., 2015a). Due to less high rotating components, the two-stage gearbox is more reliable than both three stage versions. Furthermore, the exchange of a two-stage gearbox is less expensive as the gearbox is lighter and has lower investment cost. A distinction between three stage gearboxes
with a three point and a four-point suspension is discernible. Due to the non-torque loads entering the gearbox with a three-

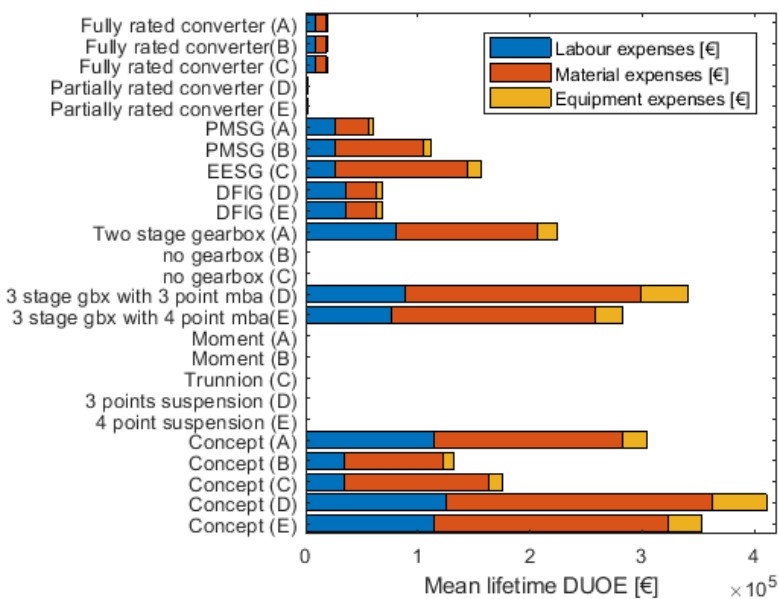

**Figure 5: Drivetrain concept comparison based on 1,000,000 iterations for a 3 MW and 120 m rotor diameter design**

point suspension system, it is less reliable and leads to higher unplanned operational effort. Going further to the component generator it is visible, that the EESG leads to the highest mean DUOE. Reliability wise PMSG and EESG seem to be on the same level. This is derived from the same labour expenses level and Weibull parameters. Still material expenses are higher for the EESG as its investment cost and therefore material expenses are higher. It is furthermore, heavier than the direct drive PMSG resulting in higher equipment expenses. Despite its higher failure rates, the DFIG results in lower mean DUOE than the direct drive synchronous generators. As the DFIG combined with a three-stage gearbox operates in higher rotational input speed ranges and lower rated input torque, it needs fewer pole pairs. This leads to a comparatively less complex, lighter and less expensive generator for the considered rated power and rotor diameter. The same argumentation is valid when comparing the results for geared and direct drive PMSG (A vs. B). Due to the higher rotational input speed and lower input torque, the generator needs fewer pole pairs and a less stiff structure. Looking into the behaviour of the component converter it is visible, that the converter has a minor influence on the overall expenses. The reason is the low amount of needed replacements which usually lead to high expenses. This is in line with literature which says that converter failures can often be solved remotely or with low effort.

Table 4 shows the concept specific DUOEs lifetime impact on the total drivetrain lifetime effort. In this approach the total drivetrain lifetime effort consists of the mean lifetime DOUE and the calculated drivetrain specific investment cost. The latter is based on calculations from NRELs Drivetrain Cost and Scaling model (Fingersh et al., 2006). Logistics and installation effort are not included as it cannot directly be assigned to the drivetrain. In the literature it is usually assigned to the entire turbine. Based on the underlying assumptions concept B seems to be the dominant concept in terms of total drivetrain lifetime

**Table 4: Concept specific DUOEs lifetime impact on the total drivetrain lifetime effort for the 3 MW 120 m rotor application**

| Concept | A | B | C | D | E |
|---|---|---|---|---|---|
| 3 MW and 120 m rotor diameter application | | | | | |
| Mean lifetime DUOE [€] | 305,160 | 131,620 | 174,800 | 410,610 | 354,270 |
| Calculated drivetrain specific investment cost [€] | 748,800 | 874,700 | 1,143,700 | 861,300 | 861,300 |
| Total drivetrain lifetime effort [€] | 1,053,960 | 1,006,320 | 1,318,500 | 1,271,910 | 1,215,570 |
| Share of mean lifetime DUOE on total drivetrain lifetime effort [%] | 28.95 | 13.08 | 13.26 | 32.28 | 29.14 |

effort for the considered application. Finally, the concept specific mean lifetime DUOE share on the total drivetrain lifetime effort is presented. Based on the model results it is discernible that for geared drivetrains (concept A, D and E) the specific DUOE can account for up to a third of the total drivetrain lifetime effort. For direct versions (concept B and C), it is around

13 percent.

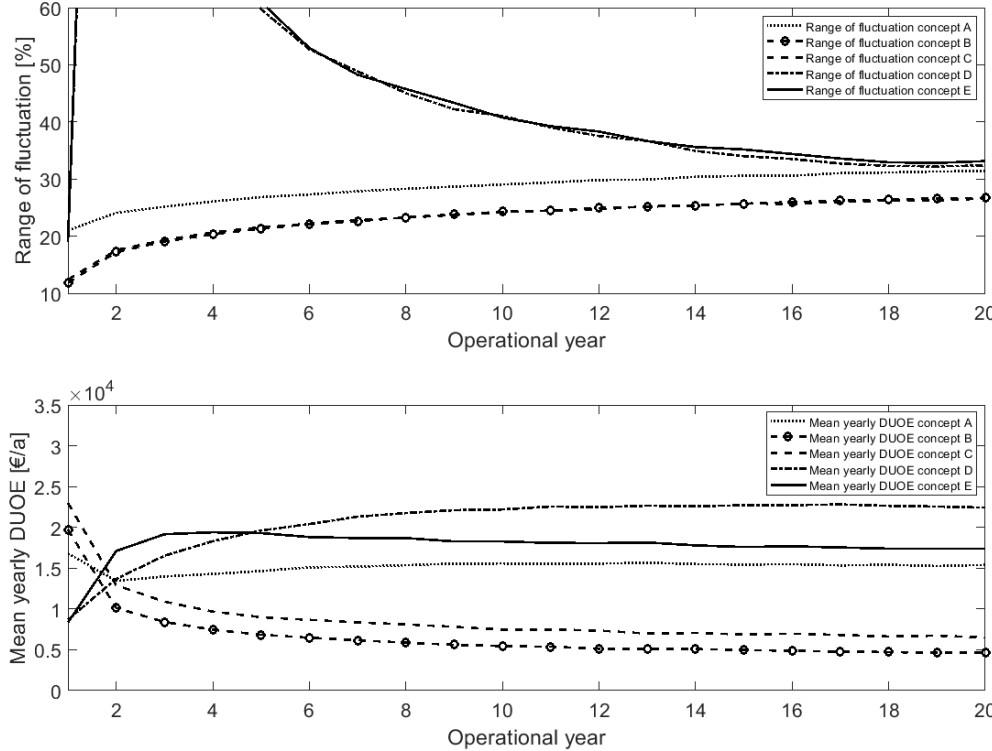

**Figure 6: Mean unplanned yearly DUOE and yearly range of fluctuation of DUOE for different drivetrain concepts based on 1,000,000 iterations for a 3 MW 120 m rotor application**

The bottom plot of Figure 6 gives an overview about the mean DUOE of the drivetrain concepts and its development over the lifetime. In order to explain the course of the graph in Figure 6 the combination of the failure behaviour of each component in the concept specific design should be kept in mind as it has a high influence on DUOE. Please recall, Table 2 gives an overview about the modelled Weibull parameters for failure behaviour of different drivetrain components. Concept A, D and E all show

a dominant wear out behaviour (high level of DUOE at the end of the lifetime) which can be traced back to the used three

stage gearbox. Furthermore, concept D and E both have a DFIG, which additionally leads to the shown wear out behaviour. Interestingly random failure behaviour is visible for these three concepts, as the DUOE stays at a constant level after its infancy. Concept A, B and C all use synchronous generators which all follow mainly an early failure behaviour. This is visible in the course of the graph in Figure 6, as it starts at a high level and decreases within the first years of operation. As visible in Figure 5 the chosen converter concept has a minor impact onto the mean modelled lifetime DUOE. Accordingly, the impact on the

course of Figure 6 is negligible.

Still mean values do not allow a statement about the results certainty. In order to allow a statement about the certainty of this behaviour the range of fluctuation is calculated for a worst-case scenario. The range of fluctuation is defined as the concept's individual yearly standard deviation of DUOE divided by the mean concept's DUOE c.f. Formula (3):

$$range\ of\ fluctuation(i) = \sum_{j=1}^{4} \frac{\sigma_{j,d}(i)}{\mu_{j,d}(i)} \tag{3}$$

In Formula (3) $\mu_{j,d}(i)$ depicts the mean DOUE of component $j$ with design $d$ in respective year $i$. Whereas, $\sigma_{j,d}(i)$ depicts the

standard deviation of DUOE for the component $j$ in design $d$ under investigation in respective year $i$. Worst case is defined, as the sum of fluctuation of the individual components in the design under investigation, which though will unlikely come into effect. Range of fluctuation is an indicator of inherent risk as it gives an indication about the possible maximum deviation from the mean. For better vividness, the plot is cut at a range of fluctuation of 60. From an inherent risk point of view, the direct drive concepts perform best, please compare the top plot of Figure 6. Still worst case can be a maximum deviation of 10 – 30

times to the yearly mean DUOE. Risk of deviation rises until the end of the drivetrain's lifetime. The two three stage gearbox concepts perform worse from a risk inherent perspective. Especially concept D can have 560 times the mean yearly DUOE in a worst-case scenario in the early lifetime. Concept E can result in expenses over 300 times the mean yearly value. It needs to be kept in mind that the failure behaviour for gearboxes is derived from a lot more data points than the behaviour of the other components. This can lead to a higher deviation as more possible applications are covered. A solely technical cause is

questionable.

In order to bring the above presented results into perspective the historical input data is questioned. Experts have been consulted in order to evaluate if the used data still presents todays' technology behaviour. It turns out, that wind turbine OEMs on average calculate with one gearbox exchange over the turbine's lifetime. The used historical data sample indicates less reliable behaviour. One development which is not reflected in the data is the improved planned maintenance of gearboxes which

improved today's gearboxes' reliability. This has been mentioned as a lesson learned from previous experience. Furthermore, gearbox OEMs more and more design gearboxes which can be repaired up tower. This development makes it possible to reduce the amount of major gearbox replacements and have major repair instead. This effect is not captured in the historical data yet.

Besides evaluating todays' technology, it is even more interesting to have a look at possible future developments. Though the authors are not able to anticipate future development with certainty they can utilize the presented method to give indications about possible trends. Figure 7 presents results about future development based on assumptions whose grounding is experience and developments already visible today. The application for future turbines is characterized by a rated power of 5 MW and a rotor diameter of 150 m, being the average wind OEMs announced in 2020 for onshore application (Bundesverband Wind Energie, 2020).

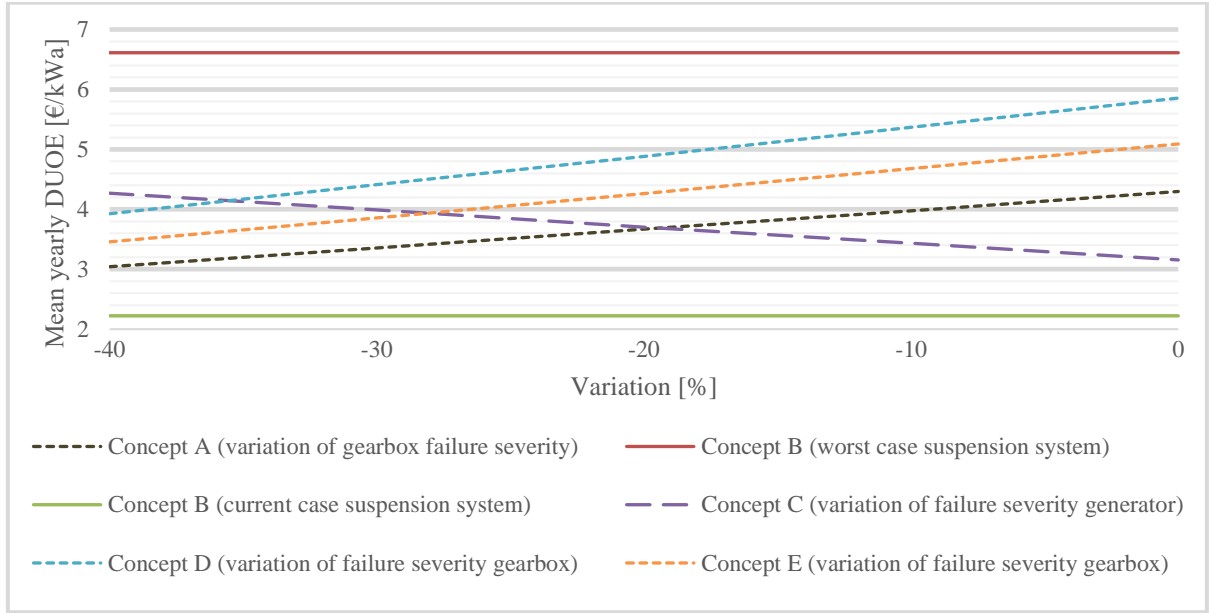

**Figure 7: Impact of future development on the yearly mean DUOE for a 5 MW 150 m rotor diameter onshore application**

A current trend in gearbox development is an increased torque density especially for higher rated power applications. Higher power density allows to use smaller gears which lead to smaller and lighter gearboxes. More compact gears are less exposed to material imperfections and therefore a smaller probability of fatigue failures exists. It is assumed, that exact calculation methods for the strength against the predominant failure modes of the given gear sets are available as downsizing leads to a reduction of the remaining safety reserves and this effect could also lead to reduced reliability. In order to enable higher torque densities plain bearings are seen as an enabler for more compact gearbox design. Introducing this new bearing design might decrease reliability. Combining all mentioned factors, it is assumed, that future gearbox reliability behaviour stays constant but corrected in terms of the above-mentioned reduced amount of major gearbox replacements due to the age of the input data. A sensitivity study varying the amount of major replacements and having major repair instead for concept A, D and E is shown in Figure 7.

For concept C a reliability deterioration for the generator is expected. The turbine OEM currently using this concept introduced a technology shift in the coil material and manufacturing in order to decrease the investment cost. The wires are welded together and aluminium is used instead of copper (Enercon GmbH). First of all, experience needs to be gained with this new

technology in combination with meeting increased loading requirements (higher input torques). Therefore, failure severity might increase for the generator of concept C in Figure 7.

For the concepts with a moment bearing a reduced reliability is possible for the future. Unlike the other bearing concepts this suspension concept is comprised in one point of support, which leads to a disproportionate increase of dimensions if input loads rise. This makes it likely to switch to plain bearings. Moreover, this concept needs high manufacturing and mounting accuracy, which is more difficult to meet if dimensions increase. Both aspects make it likely to have reduced reliability. As the main bearing arrangement failure occurrence was negligible in the former analysis a best and a worst case are plotted in Figure 7. Best case is defined as a suspension system reliability similar to the historically reported. Worst case is defined by a triangulation with a mean time to failure of 10 years (min and max: 0 – 20 a). Furthermore, all reported failures are interpreted as major replacement. This assumption is based on expert interview.

Figure 7 depicts possible future technological developments and their impact on the mean yearly DUOE of different drivetrain concepts. A variation of 0 percent depicts the mean yearly DUOE based on the same assumptions as the results for the previously presented 3 MW application. It is visible, that the order of advantageousness is not changed due to the change in the application requirements. Though it is furthermore visible that, at certain combinations of developments the order of advantageousness of the concepts is changed. It is up to the reader to assess how realistic these developments are. Concluding from Figure 7, the superiority of synchronous generator concepts manifested in historic data is not entirely certain in future application. It mostly depends on a myriad of developments in the different technologies.

Another approach to derive future drivetrains reliability behaviour introduced by Moghadam et al. 2020 identifies positive aspects of a geared drivetrain concepts which might be dominant in certain applications (Moghadam and Nejad, 2020). They conduct a reliability comparison for future drivetrain concepts for a 10 MW offshore floating application. By simulating rotor torque and generator electromagnetic torque oscillation for different concepts and argumentatively analyzing the influence on the drivetrains failure modes, they identify several effects of a mid or high speed concept using a gearbox which probably reduce the coincidence of frequencies and vibration. This way they argue geared concepts might lead to higher reliability in that application if new failure modes due to the presence of a gearbox do not mitigate them. They furthermore take the entire lifecycle into consideration and see further advantages (weight, investment cost and efficiency) of using a geared concept.

Conclusion and Outlook

In order to identify holistically better drivetrain concepts for onshore application, their operational behaviour needs to be taken into account in an early design phase. In this paper, a validated approach for estimating drivetrain concept specific risk of unplanned maintenance effort and technical availability based on open access data is presented. By describing uncertain influencing factors with distributions, the poor data availability in literature and in the early design phase can be used to get an indication about the concepts choice influence on the unplanned operational turbine behaviour. This approach furthermore allows to include information about the concept's behaviour from different applications and sources. By using triangulation incremental innovation and completely new concept ideas can be evaluated as well. In order to get representative comparisons a Monte Carlo method is applied. This way a multitude of drivetrain lifetimes can be modelled following the distributions

behaviour. The most relevant influencing factors are considered by modelling failure rate, downtime, failure severity and duration of repair and replacement as uncertain factors. Technical availability and drivetrain influenced unplanned maintenance effort are defined as evaluation criteria. By calculating the range of fluctuation of the results, this approach offers an indication about the inherent risk in the drivetrain influenced unplanned maintenance effort which is a central criterion. Scalability is given, as material and equipment expenses are scaled with turbine rotor diameter and rated power. This approach shows that openly accessible data or expert estimations are sufficient for comparing different drivetrain concepts over the operational phase in an early design stage.

The application of this approach on five state-of-the-art drivetrain concepts for a 3 MW, 120 m rotor diameter turbine demonstrates that for all concepts and components the material expenses have the highest influence on mean DUOE followed by labour expenses. Equipment expenses, if modelled in the way presented, are less influential. Overall direct drive concepts lead to the lowest mean DUOE over the lifetime. This indication is confirmed when looking on the inherent risk of deviations from these estimated mean values. A gaze into the future showed, that the superiority of synchronous generator concepts manifested in historic data is not entirely certain in future application. Exemplary the impact of higher torque density in gearboxes, a change to moment bearings and adjusted coil design in electrically excited generators have been considered.

Still it has to be considered, that this analysis is based on sometimes very old and maybe outdated data especially when describing the failure behaviour. Furthermore, the extent of the databases for different component design deviates a lot which might bias the result. Unfortunately, a component design specific distinction of the failure severity is not possible based on open access data up to now. For adapting this method to new concepts, a physically based approach could be developed which would make it possible to estimate probability distributions for the uncertain factors. In the presented approach no specific failure mechanisms are discerned, they are aggregate into three types. For future research it could be a highly interesting topic to derive failure mechanism specific Weibull distributions. Moreover, dependency between failures of different components is only indirectly taken into account, this could be addressed additionally. Another possible direction for research is to include the influence of the maintenance strategy as well as site or park specific impacts in the evaluation. Moreover, this approach only takes the operational phase into account. For identifying holistically superior drivetrain concepts, the entire drivetrain lifecycle needs to be considered. The authors intend to develop approaches for estimating the concepts behaviour in all lifecycle phases of the drivetrain which can deal with the poor data availability in the early development phase. This way they will be able to evaluate different trade-offs within the drivetrain design. Nevertheless, this approach can already assist in drivetrain concept decision making by being able to quantify the inherent technological risks in the operational phase.

### Data availability

The data and metadata used to calculate the results shown in this study are cited and accessible via URL in the bibliography. The datasets are not yet deposited in a DOI marked repository because there are still substantial adaptions being made to the described models.

## Author contributions

Freia Harzendorf did the conceptualization, data curation, formal analysis, methodology development, validation, writing and editing of this paper. Funding acquisition has been done by Prof. Schelenz and Prof. Jacobs.

### Competing interests

The authors declare that they have no conflict of interest.

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
