# Peer review of "Reducing cost uncertainty in the drivetrain design decision with a focus on the operational phase"

_Wind Energy Science, 2020_

## Referee Comment (RC1) · Amir R. Nejad (Referee) · 12 Apr 2020

This article compares the unplanned maintenance cost of 5 different drivetrain technologies for 3 MW, land-based turbines. The paper is well written and topic is of interest. Here are some specific comments and suggestions:

-The comparison will be more interesting if the cost of unplanned maintenance together with the probability of its occurrence are presented. Please comment on the probability of occurrence of unplanned maintenance for each concept over the life span.

-It would be interesting to mention the share of such operational cost in the total cost

(what portion of OPEX and what portion of CAPEX+OPEX). What is the CAPEX cost for each drivetrain concept? Literature review can be extended, in particular looking at relevant literature addressing total cost of drivetrain or over the life cycle (not only operation) - see for instance this https://doi.org/10.1002/we.2499 for drivetrains on offshore turbines.

-Fig. 4: please comment on market share each concept has and what are their CAPEX estimates.

-Fig 6: there is a point between year 2-4 where the mean value becomes almost steady and constant for most of the concepts, please provide explanation. It would be nice to have the mean and standard deviation figures together.

-Page 14: "Despite the higher effort for their generator and converter designs they are superior as they can operate without a gearbox". "As the EESG investment is more expensive and heavier than the PMSG for the same application, a direct drive with a PMSG is the winner in this comparison." Comprehensive comparison of different designs is not the scope of this paper, therefore such these statements seems to be too general.

-There are some typos, e.g. in duction instead of induction, and some grammar mistakes which needed more careful proofreading.

-The definition of variables in eqs must be improved. The variable (a) is not defined in eq. 1. It is also not explained why s, j and d are taking those values. I assume "a" in figures 4 and 6 refers to annual.

-It would be interesting if the authors could comment the same study on different power ranges.

-The concept stated by the statement "Having included component specific mass and cost makes this approach scalable in rated power and rotor diameter." In lines 137 and 138 needs more elaboration and justification.

---

## Referee Comment (RC2) · Anonymous Referee #2 · 16 Oct 2020

The paper addresses an important issue, and, in my view, should be published as an interesting contribution to an ongoing discussion. However, it cannot be regarded as excellent or a major breakthrough. The conclusions are a bit too trivial. They reveal that the proposed method is in principle working, but still suffering from a lack of reliable accessible data. So the original problem is not yet solved. And the advantage of direct drives versus drives with gears when using such a methodology in the way described in the paper comes as no surprise at all. What is not considered is e.g. the role of power density. Gears mean higher power density, and therefore less volume of material. And therefore, less risk of material imperfections and smaller probability of fatigue failures originating from such flaws. This aspect should be addressed.

In detail, I have a remark regarding line 240:

"...whereas the two-stage gearbox, the three-stagegearbox with a three-pointsuspension system, and the DFIG can mainly be attributed to wear out behavior" The term "wear out behavior" is too unspecific and incomplete. There are numerous failure modes and among them are fatigue and also wear; the expression "wear" in itself is comprising various mechanisms. The suitable distributions, e.g. Weibull, and their shapes vary considerably. Therefore, I regard the way gearbox failures are considered summarily and indiscriminate as "wear out" as too simplistic. There should be a comment on this!

---

## Author Comment (AC1) · 12 Nov 2020

Dear respected reviewer,

Thanks a lot for the helpful remarks on the manuscript. For a faster understanding of the changes made I allowed myself to copy the comments and type my response and changes made in the manuscript below, R stands for the referee and A for the authors answers.

R: This article compares the unplanned maintenance cost of 5 different drivetrain technologies for 3 MW, land-based turbines. The paper is well written and topic is of inter-

[Figure]

est. Here are some specific comments and suggestions:

A: Thank you very much for this constructive review. Your comments below are specific and have highlighted place for improvement of the paper. I have tried to implement all comments in the revised version of the paper.

R: The comparison will be more interesting if the cost of unplanned maintenance together with the probability of its occurrence are presented. Please comment on the probability of occurrence of unplanned maintenance for each concept over the life span.

A: In the authors eyes the presented expected value of drivetrain influenced unplanned operational effort is adding more value to the discussion than the probability of occurrence of unplanned maintenance. As the first incorporates the impact failure rate, failure severity as well as repair and downtime as uncertain factors have in combination on the result. Whereas unplanned maintenance indicates a happening of a failure but not the consequences this entails effort wise. The presented approach uses distribution functions which are based on literature values cited in the table in the bottom of Figure 2. Based on the distribution functions for failure rate respective mean time to failure it is possible to derive the probability of occurrence of unplanned maintenance for each concept. A table with the used literature values for deriving the distribution functions can be added into the supplementary material if requested.

R: It would be interesting to mention the share of such operational cost in the total cost (what portion of OPEX and what portion of CAPEX+OPEX). What is the CAPEX cost for each drivetrain concept?

A: Indeed, it would be interesting to see the share of unplanned operational drivetrain effort on the total cost of the drivetrain over the lifetime. Therefore, Table 4 (uploaded and see p. 12 in the revised manuscript) has been added showing the concept specific mean unplanned drivetrain influenced operational lifetime effort and the concept specific mean drivetrain component investment effort both in euro. The drivetrain component specific investment effort in euro is based on calculations from NRELs Drivetrain

Cost and Scaling model. Logistics and installation effort is not included as it cannot directly be assigned to the drivetrain. In literature it is usually assigned to the entire turbine. Finally the concept specific mean unplanned drivetrain influenced operational lifetime effort share on the total drivetrain lifetime effort is presented in the bottom of Table 4.

R: Literature review can be extended, in particular looking at relevant literature addressing total cost of drivetrain or over the life cycle (not only operation) - see for instance this https://doi.org/10.1002/we.2499 for drivetrains on offshore turbines.

A: The author intentionally limited the scope of this paper to the comparison on wind turbine drivetrain concepts with regards to unplanned operational drivetrain influenced effort. This uncertain topic has a high degree of complexity and needs a thorough analysis. Therefore, no literature addressing the total cost of drivetrain over the life cycle is added in the literature review. The suggested paper by the reviewer is well written, nevertheless it has a completely different focus, dealing with 10 MW offshore drivetrain technologies. Furthermore, it seems a bit strange, that the reviewer suggests a paper written by himself. Nevertheless, the authors did another literature research in the databases 'web of science' and 'scorpus'. They used the keywords "drivetrain" and "wind" in combination. Available literature analysing the drivetrain operational phase mostly deal with condition monitoring or very specific failure mechanisms as well as with different types of loading and dynamics which are not in the scope of this paper. Therefore, no further literature has been added.

R: Fig. 4: please comment on market share each concept has and what are their CAPEX estimates.

A: A source stating the market share of each drivetrain concept has been added to the table in Figure 4 (see p. 9). CAPEX estimates are included in Table 4, as stated in the comment above.

R: Fig 6: there is a point between year 2-4 where the mean value becomes almost

steady and constant for most of the concepts, please provide explanation.

A: In order to explain the course of the graph in Figure 6 the combination of the failure behaviour of each component in the concept specific design should be kept in mind. Table 2 gives an overview about the modelled Weibull parameter for the failure behaviour of different drivetrain components. Concept A, D and E all show wear out behaviour (high level of mean unplanned operational effort at the end of the lifetime) which can be traced back to the used three stage gearbox. Furthermore, concept D and E both have a DFIG generator which also leads to the shown wear out behaviour. Interestingly random failure behavior is visible for these three concepts, as the DUOE stays at a constant level after its infancy. This is the authors explanation for the almost steady and constant course after year 2 – 4. Concept A, B and C all use synchronous generators where early failure behaviour is dominant. This is visible in the course of the graph in Figure 6, as it starts at a high level and decreases within the first years of operation. As visible in Figure 5 the chosen converter concept has a minor impact onto the mean unplanned drivetrain influenced operational lifetime effort modelled. Accordingly the impact on the course of Figure 6 is also negligible. This explanation is added in the paper.

R: It would be nice to have the mean and standard deviation figures together.

A: For better vividness the authors decided to plot mean and standard deviation indirectly in a figure for the yearly development (adjusted Figure 6 - uploaded). The introduced metric 'range of fluctuation' incorporates mean and standard deviation.

R: Page 14: "Despite the higher effort for their generator and converter designs they are superior as they can operate without a gearbox". "As the EESG investment is more expensive and heavier than the PMSG for the same application, a direct drive with a PMSG is the winner in this comparison." Comprehensive comparison of different designs is not the scope of this paper, therefore such these statements seems to be too general.

A: The reviewer is totally right. The paragraph is changed in the following way: "The application of this approach on five state-of-the-art drivetrain concepts for a 3 MW, 120 m rotor diameter turbine shows that for all concepts and components the material expenses have the highest influence on mean DUOE followed by labour expenses. Equipment expenses, if modelled in the way presented, are less influential. Overall direct drive concepts lead to the lowest mean DUOE over the lifetime. This indication is confirmed when looking on the inherent risk of deviations from these estimated mean values."

R: There are some typos, e.g. in duction instead of induction, and some grammar mistakes which needed more careful proofreading.

A: The grammar and typos in the paper were revised. Please excuse the previous mistakes. R: The definition of variables in eqs must be improved. The variable (a) is not defined in eq. 1. It is also not explained why s, j and d are taking those values. I assume "a" in figures 4 and 6 refers to annual.

A: Thanks to the hint of the reviewer Formula (1) has been changed and includes now a variable h which indicates the number of hours a calendar year has. Furthermore, examples for s, j and d are given for improving the understandability of these variables.

R: It would be interesting if the authors could comment the same study on different power ranges.

A: Thanks to the reviewers comment the manuscript is now expanded to a look at possible future developments. The application for future turbines is characterized by a rated power of 5 MW and a rotor diameter of 150 m, being the average wind OEMs announced in 2020 for onshore application. The order of advantageousness is not changed due to the change in the application requirements, without taking technological development into account. Though the authors are not able to anticipate future development with certainty they can utilize the presented method to give indications about possible trends. Exemplarily the possible impact of higher torque density in gearboxes,

a change to moment bearings and adjusted coil design in electrically excited genera-tors is incorporated. It shows, that the superiority of synchronous generator concepts manifested in historic data is not entirely certain in future application.

R: The concept stated by the statement "Having included component specific mass and cost makes this approach scalable in rated power and rotor diameter." In lines 137 and 138 needs more elaboration and justification.

A: More elaboration and justification is added in line 148 ff on page 5 of the manuscript. Both component design specific weight and component design specific investment cost scale with rated power and rotor diameter and therefore with the application. They are calculated based on the NREL Cost and Scaling Model (Fingersh et al., 2006), which is a cost and mass regression model based on industry data. As visible in Formula (2) these two variables have an impact on the material expenses as well as on the equipment expenses and though a high impact on DUOE. This way the use of these inputs makes this approach scalable in rated power and rotor diameter.

I hope all remarks are taken into account to your satisfaction.

With best regards, Freia Harzendorf

————————————————————

[Figure]

**Fig. 1.** Figure 6 from the manuscript: Mean unplanned yearly DUOE and yearly range of fluctuation of DUOE for different drivetrain concepts based on 1,000,000 iterations for a 3 MW 120 m rotor application

**Table 1: Concept specific DUOEs lifetime impact on the total drivetrain lifetime effort for the 3 MW 120 m rotor application**

| Concept | A | B | C | D | E |
|---|---|---|---|---|---|
| **3 MW and 120 m rotor diameter application** | | | | | |
| Mean lifetime DUOE [€] | 305,160 | 131,620 | 174,800 | 410,610 | 354,270 |
| Calculated drivetrain specific investment cost [€] | 748,800 | 874,700 | 1,143,700 | 861,300 | 861,300 |
| Total drivetrain lifetime effort [€] | 1,053,960 | 1,006,320 | 1,318,500 | 1,271,910 | 1,215,570 |
| Share of mean lifetime DUOE on total drivetrain lifetime effort [%] | 28.95 | 13.08 | 13.26 | 32.28 | 29.14 |

**Fig. 2.** Table 4: Concept specific DUOEs lifetime impact on the total drivetrain lifetime effort for the 3 MW 120 m rotor application

---

## Author Comment (AC2) · 12 Nov 2020

Dear respected reviewer,

Thanks a lot for the helpful remarks on the manuscript. For a faster understanding of the changes made I allowed myself to copy the comments and type my response and changes made in the manuscript below, R stands for the referee and A for the authors answers.

R: The paper addresses an important issue, and, in my view, should be published as an interesting contribution to an ongoing discussion. However, it cannot be regarded

as excellent or a major breakthrough. The conclusions are a bit too trivial.

A: Thanks to your comments, the conclusion section has been rewritten in order to point out the important findings which give valuable insights for the above mentioned discussion. Furthermore, some more application fields and sensitivity analysis have been added to the concept comparison section in order to show the approaches value.

R: They reveal that the proposed method is in principle working, but still suffering from a lack of reliable accessible data. So the original problem is not yet solved.

A: The authors presented a framework for analysing the unplanned maintenance effort due to the drivetrain concept and already filled it with the available historical data. From a statistical point of view the filled data is sufficient enough to allow drawing conclusions. Nevertheless, this framework showed it workability and can now be used for sensitivity analysis and the testing of the influence of future development on this key performance indicator. The referee nevertheless is right that the original problem is not entirely solved. We therefore suggest to develop a physically based approach which would make it possible to estimate probability distributions for the uncertain model input factors.

R: And the advantage of direct drives versus drives with gears when using such a methodology in the way described in the paper comes as no surprise at all. What is not considered is e.g. the role of power density. Gears mean higher power density, and therefore less volume of material. And therefore, less risk of material imperfections and smaller probability of fatigue failures originating from such flaws. This aspect should be addressed.

A: Your comment inspired to add two paragraphs to the manuscript. The first paragraph deals with the question, if the historical data still presents todays' technologies behavior. It comes to the conclusion, that current development should be reflected in an reduced amount of major gearbox replacements and an increased number of major repairs instead. The second paragraph gives insight on how the different drivetrain

concepts might perform in future application in terms of drivetrain influenced unplanned operational effort. The application for future turbines is characterized by a rated power of 5 MW and a rotor diameter of 150 m, being the average wind OEMs announced in 2020 for onshore application. The order of advantageousness is not changed due to the change in the application requirements, without taking technological development into account. Though the authors are not able to anticipate future development with certainty they can utilize the presented method to give indications about possible trends. Exemplarily the possible impact of higher torque density in gearboxes, a change to moment bearings and adjusted coil design in electrically excited generators is incorporated. It shows, that the superiority of synchronous generator concepts manifested in historic data is not entirely certain in future application.

R: In detail, I have a remark regarding line 240:"...whereas the two-stage gearbox, the three-stage gearbox with a three-point suspension system, and the DFIG can mainly be attributed to wear out behavior" The term "wear out behavior" is too unspecific and incomplete. There are numerous failure modes and among them are fatigue and also wear; the expression "wear" in itself is comprising various mechanisms. The suitable distributions, e.g. Weibull, and their shapes vary considerably. Therefore, I regard the way gearbox failures are considered summarily and indiscriminate as "wear out" as too simplistic. There should be a comment on this!

A: The referee is totally right. Gearboxes and all other drivetrain components can fail due to a variety of failure modes, which in combination lead to the failure rates available in literature. Due to a lack of failure mechanism specific failure rates in literature, the concept of the bathtub curve is applied here. In the context of bathtub curve wear out is understood in the general way of abrasion and comprises various mechanisms which lead to a failure in the end of a components lifetime. From the used data sources which are studies including aggregated information about failure rates of different drivetrain components for fleets of turbines it is not comprehensible which failure mechanisms lead to the breakdown. Nevertheless, this simplistic distinction is sufficient for giving

preliminary statements on the unplanned operational effort in the design stage of a drivetrain concept. Sure, this approach can also be utilized for analysing the effect of a special failure mechanism on the drivetrain induced operational effort. For being able to conduct this analysis a failure mechanism specific distribution function is needed. In the end this analysis can give insights on how expensive the further development of a component should be in order to prevent this specific failure. Nevertheless, this was not the focus of the papers considerations.

I hope all remarks are taken into account to your satisfaction.

With best regards,

Freia Harzendorf
* * *